# Long-Term Usage of Proton Pump Inhibitors Associated with Prognosis in Patients with Colorectal Cancer

**DOI:** 10.3390/cancers15215304

**Published:** 2023-11-06

**Authors:** Chin-Chia Wu, Chuan-Yin Fang, Ben-Hui Yu, Chun-Ming Chang, Ta-Wen Hsu, Chung-Lin Hung, Shih-Kai Hung, Wen-Yen Chiou, Jui-Hsiu Tsai

**Affiliations:** 1Division of Colorectal Surgery, Buddhist Tzu Chi Medical Foundation, Dalin Tzu Chi Hospital, Chiayi 622, Taiwan; jimmywcc@tzuchi.com.tw (C.-C.W.); b120018@tzuchi.com.tw (T.-W.H.); 2School of Post-Baccalaureate Chinese Medicine, Tzu Chi University, Hualien 970374, Taiwan; 3Division of Colon and Rectal Surgery, Ditmanson Medical Foundation Chia-Yi Christian Hospital, Chiayi 600566, Taiwan; 04969@cych.org.tw; 4Department of Radiation Oncology, Buddhist Tzu Chi Medical Foundation, Dalin Tzu Chi Hospital, Chiayi 622, Taiwan; bhyu0418@gmail.com (B.-H.Y.); oncology158@tzuchi.com.tw (S.-K.H.); 5Department of General Surgery, Buddhist Tzu Chi Medical Foundation, Hualien Tzu Chi Hospital, Hualien 970, Taiwan; ccmjim@tzuchi.com.tw; 6School of Medicine, Tzu Chi University, Hualien 970, Taiwan; 7Department of Oncology, Buddhist Tzu Chi Medical Foundation, Dalin Tzu Chi Hospital, Chiayi 622, Taiwan; dl09454@tzuchi.com.tw; 8Department of Psychiatry, Buddhist Tzu Chi Medical Foundation, Dalin Tzu Chi Hospital, Chiayi 622, Taiwan

**Keywords:** colorectal cancer, proton pump inhibitor, dose response, mortality, recurrence

## Abstract

**Simple Summary:**

Proton pump inhibitors (PPIs) are commonly used for acid reduction in peptic ulcer diseases and esophageal reflux diseases. The effects of changing gut microbiota, and prolonged hypergastrinemia were reported. This study demonstrated that long-term PPI use in CRC patients was associated with an increased risk of death with dose–response effect but not for recurrence. PPIs may play a complex role in CRC patients.

**Abstract:**

The dose–response effect of proton pump inhibitors on colorectal cancer prognosis is still under exploration. This population-based study in Taiwan was designed to examine the effect of proton pump inhibitors on overall death, colorectal cancer-specific death, and recurrence in colorectal cancer patients with different cumulative proton pump inhibitor dose levels. This cohort study was based on the Taiwan Cancer Registry and Taiwan National Health Insurance Research Database from 2005 to 2020. After frequency matching with a 1:1 ratio, a total of 20,889 users with proton pump inhibitors and 20,889 without proton pump inhibitors were analyzed. The cumulative defined daily dose level of proton pump inhibitor was stratified to explore the dose–response relationship. A proton pump inhibitor exposure cumulative defined daily dose > 60 after colorectal cancer diagnosis had higher risk of all-cause death than non-proton pump inhibitor users with adjusted hazard ratios of 1.10 (95% CIs: 1.04–1.18). For recurrence, a proton pump inhibitor exposure cumulative defined daily dose > 60 had reduced recurrence risk with an adjusted hazard ratio of 0.84 (95% CIs: 0.76–0.93). This study demonstrated that the long-term use of proton pump inhibitors in patients with colorectal cancer was associated with an increased risk of death that related to the proton pump inhibitor exposure cumulative defined daily dose > 60 and had different dose–response effect in various dose level.

## 1. Introduction

Colorectal cancer (CRC) is one of the most important cancers globally [1]. CRC has the second highest incidence in Taiwan and is responsible for the third most deaths in Taiwan [2]. Cancer prognosis is not only related to the cancer itself, but also the host and microenvironments [3]. Alterations to the microenvironment influence the cancer behavior and patient outcomes, including diet [4], medication [5], and microbiota [6]. 

Several drugs have been identified as having the potential to reduce CRC development and improve the prognosis, such as metformin [7], statins [8], or aspirin [9]; conversely, the association between long-term use of proton pump inhibitors (PPIs) and increased CRC risk has also been reported [10]. Therefore, CRC patients having a more severe cancer status might lead to increased consumption of analgesics (including NSAIDs), thereby, more PPIs might be prescribed. PPIs targeting gastric H(+)/K(+)-ATPase (ATP4) and inhibit the gastric acid are the most effective available therapeutic drugs for acid-related diseases [11] such as peptic ulcers, gastrointestinal bleeding, gastroesophageal reflux, and the eradication of the Helicobacter pylori [12]. The long-term use of PPIs and acid reduction are reported to be associated with subsequent pneumonia [13] and CRC development [14]. 

The literature has reported the effects of long-term use of PPIs. For example, a chronic state of hypergastrinemia induced by feedback process between gastric acid and serum gastrin [15] has been reported to be associated with an increased risk of colorectal cancer [16]. Previous epidemiological studies have indicated an association between PPI use and increased CRC risk [17]. In a study on Taiwanese population, the association between the increased dosage and increased incidence of CRC was reported [18]. However, the influence of PPIs on CRC was still controversial. A meta-analysis in 2020 reported that there was no statistically significant association between the use of PPIs and the risk of CRC [14]. In CRC patients, the effects of PPIs have also been explored, but are still not fully understood. The PPI omeprazole has been reported to have a synergetic effect on improving the effect of chemoradiotherapy and decreasing rectal cancer recurrence [19]. In contrast, a retrospective observational study in Japan, that included 606 patients between 2009–2014, investigated the association of PPIs and relapse-free and overall survival in stage 2–3 CRC patients treated with capecitabine monotherapy and a CapeOX regimen. It was reported that the co-administration of PPIs with either regimen was associated with poorer relapse-free survival and overall survival [20]. 

Results from studies investigating the association between PPI use and the prognoses of CRC patients are not consistent in terms of survival and recurrence. The current matched cohort study, based on data from the Taiwan National Health Insurance Research Database (NHIRD), was designed to explore the hypothesis that the use of PPIs could be associated with alterations in the prognosis of CRC patients. We examined the association between PPI use and CRC prognoses of all-cause death, cancer-specific death, and recurrence using medical records from Taiwan’s whole population. 

## 2. Materials and Methods 

### 2.1. Ethics Statement 

This research was performed in accordance with the relevant guidelines and regulations [21,22]. All identifiable personal information was removed by the Taiwan National Health Insurance (NHI) program from the initial dataset prior to analysis. 

De-identification ensures that any medical records are unable to be tracked down to any individual patients; hence, informed consent was not required, and this study was approved by the Institutional Review Board (IRB) of the Dalin Tzu Chi Hospital (IRB approval number: B10704014). 

### 2.2. Data Source

The Taiwan NHI is a single-payer insurance system which provides universal coverage for approximately 99% of the population of Taiwan and has contracts with 97% of the medical providers [23,24]. The NHI Research Database (NHIRD) was created based on the Taiwan NHI, and the data available for this study included all medical claims made between 2005 and 2020. In addition to de-identified ID and date of visits, the medical information included diagnosis codes of diseases in ICD9CM (before 2016) and ICD10CM (since 2016), and therapy such as procedures and drugs in anatomical therapeutic chemical codes. For exploration of CRC recurrence, the Taiwan Cancer Registry (TCR) was used. TCR is a nationwide population-based registry with a high degree of accuracy in the TCR long-form data [25]. 

### 2.3. Study Population and PPI Exposure

The study design is shown in Figure 1. All colorectal cancer (CRC, ICD9CM 153–154 or ICD10CM C18–C21) patients’ data, between 2005 to 2020 in Taiwan, was obtained from the National Health Insurance Research Database and the Taiwan Cancer Registry. The comorbidities were evaluated based on clinical visits in the year prior to the first CRC diagnosis date. CRC patients diagnosed between 2006 and 2019 were incorporated. Only newly diagnosed CRC patients were incorporated into this cohort study. The index date was defined as the date of diagnosis. All patients were followed up until the year 2020 or death, with at least a one-year follow-up period (Figure 1). 

For assessment of dose–response, each patient’s cumulative defined daily dose (cDDD) of all PPIs was calculated by the World Health Organization’s recommendation [26]. To explore the effect of PPIs in new users after CRC, PPI > 60 cumulative defined daily dose (cDDD) within one year prior to CRC diagnosis were excluded to ensure sufficient accumulating exposure dosage and minimize the immortal time bias [27] (n = 7371, Figure 2). CRC patients with other cancer history (n = 36,696), or not confirmed by the catastrophic disease registration system, i.e., carcinoma in situ (n = 48,241) were excluded. Patients who died within one year after CRC diagnosis were excluded (n = 31,197). The data flowchart is shown in Figure 2. 

According to the Taiwan cancer registration rules, if patients had metastatic disease in the beginning of cancer diagnosis or if patients had persistent disease, e.g., a patient who did not receive surgery, then there would be no recurrence recording. Therefore, in this study we only included patients who received definite surgery as their primary anti-cancer treatment modality and excluded patients who did not receive surgical resection (n = 11,769, Figure 2).

The study was also designed to explore the high dose users (cDDD > 60) and low dose users (cDDD < 60) within one year after CRC diagnosis. Patients who died within one year after CRC diagnosis were excluded (n = 31,197).

### 2.4. Confounding Factors and Frequency Matching

Charlson Comorbidity Index (CCI) and individual comorbidity were evaluated from outpatient and inpatient medical claims. Cancer treatments such as chemotherapy, radiotherapy, surgery, and targeted therapy were identified within one year of a CRC diagnosis. Demographic variables such as age, sex, urbanization, region, and monthly income were based on patients’ insurance enrollment records at their first CRC diagnosis. 

An indication bias may exist while evaluating the relationship of PPI and cancer outcome. A more severe cancer status might lead to increased consumption of analgesics (including NSAIDs), thereby more PPI might be prescribed. To decrease the effect of indication bias in this study, we incorporated these variables: cancer stages, peptic ulcer disease, gastroesophageal reflux disease, gastrointestinal bleeding, steroids usage, and non-steroid anti-inflammatory drugs (NSAIDs) usage, into the frequency matching (Table 1). After matching, the above possible confounding factors had no difference between the PPI and non-PPI group (Table 1, all *p* > 0.1). Cancer site was defined by ICD9CM codes as colon left (153.2, 153.3, 153.7), colon right (153.0, 153.4, 153.5, 153.6), colon unspecified (153.8, 153.9), and rectum (154). 

After matching for possible confounding factors, a final total of 20,889 PPI users and 20,889 non-users were included in this cohort study.

### 2.5. Study Outcomes

The primary outcome was death, which was differentiated into all-cause death and cancer-specific death. Secondary outcomes were recurrence. The study was also designed to explore the effect of different cumulative PPI dosage levels. We divided the cumulative PPI dosage level within one year of CRC diagnosis into 6 groups, ≤20, 20–40, 40–60, 60–80, 80–100, and >100 cDDD. 

### 2.6. Statistical Methods

The statistical review of the study was performed with a biostatistician. Chi-squared tests were used for association in contingency tables, while Student *t*-tests were used for continuous variables. Empirical survival function was estimated using Kaplan–Meier methodologies with log-rank test, and cumulative incidence of recurrence using CIF with Gray’s modified Chi-squared test. In addition, the Kaplan–Meier methodologies with log-rank test were to estimate and compare the all-cause death in patients with colorectal cancer between PPI use and non-PPI use. Adjusted hazard ratios were estimated using Cox’s proportional hazard model. All statistical analyses were done using SAS version 9.4.

## 3. Results

Nearly half (46.8%) of the CRC patients in the current study were prescribed a PPI in their 1st year after their first CRC diagnosis. After frequency matching, PPI users and non-PPI users were comparable for each confounding factor (Table 1). Age as a discrete variable was fairly evenly distributed by cut-points at 55, 65, and 75 years, with an average age 65. Males outnumbered females by 10%. Rectal cancer accounted for 31.5%, while left-sided colon cancer and right-sided colon cancer accounted for 29.6% and 18.4%, respectively, and 20.5% was coded as unspecified. The majority of CRC was diagnosed at clinical stage III (36.9%). All the CRC patients received surgical resection prior to adjuvant treatment, followed by chemotherapy (45.1%), radiotherapy (10.9%), and targeted therapy (8.9%). 

The most common comorbidities were hypertension (32.6%) and diabetes mellitus (14.4%). Approximately 11.1% and 7.3% of patients had a history of peptic ulcer disease (PUD) and gastrointestinal (GI) bleeding, respectively. The steroids and NSAIDs use were also exactly matched in the two groups (both 65.3%). Table 1 summarizes the demographic characteristics and medical conditions in patients with colorectal cancers between PPI use and non-PPI use.

The event rates were lower in PPI users than non-PPI users in all three outcomes. However, after the duration of follow-up was taken into account, the incidence proportion (IP) were, on the opposite, higher in PPI users than non-PPI users in all three outcomes. A significant difference between high-dose users (cDDD > 60) and low-PPI-dose users (cDDD < 60) was observed in both event rates and IP (Table 2).

In Table 3, the adjusted hazard ratio (aHR) from Cox’s proportional hazard model shows the risk assessment of PPI users versus non-PPI users after taking into account all confounding factors. PPI users’ aHR (95% CI) = 1.05 (1.02, 1.09), *p* = 0.0055 for all-cause death, aHR (95% CI) = 1.04 (1.00, 1.08), *p* = 0.0436 for cancer-specific death, and aHR (95% CI) = 0.89 (0.84, 0.94), *p* < 0.0001 for recurrence.

Differences between high- and low-dose PPI was evident in overall mortality and cancer-specific death (Figure 3 and Figure 4). After adjustment, the high-dose PPI (>60 cDDD) was associated with a 10% risk increase compared to non-user, aHR (95% CI) = 1.10 (1.04, 1.18), *p* = 0.0021, while low-dose PPI (≤60 cDDD) still showed less significant difference for all-cause death (aHR (95% CI) = 1.04 (1.00–1.08), *p* = 0.0496). For cancer-specific death, the high-dose PPI (>60 cDDD) was associated with increased risk compared to non-user, aHR (95% CI) = 1.09 (1.01, 1.17), *p* = 0.021, while low-dose PPI (≤60 cDDD) was not associated with increased or reduced risk (aHR (95% CI) = 1.03 (0.99–1.08), *p* = 0.1611).

For recurrence, high-dose PPI (>60 cDDD) was also associated with a low risk compared to non-user, aHR (95% CI) = 0.84 (0.76, 0.93), *p* = 0.0012, while low-dose PPI (≤60 cDDD) was also associated with a lower risk, aHR (95% CI) = 0.90 (0.85, 0.95), *p* = 0.0003 (Table 3 and Figure 5).

### Dose–Response Relationship

Even though the majority of PPI users’ cDDD was <60 (Figure 2), to assess the dose–response relationship in a wider range, we divided the PPI users into six groups so that a polynomial regression method could be applied to estimate the relationship curve (Appendix A). The six groups defined by cDDD were ≤20 (67.4% of all PPI users), 20–40 (9.1%), 40–60 (4.3%), 60–80 (2.7%), 80–100 (3.0%), and >100 (13.4%). Subgroup analysis of Cox’s proportional hazard model estimated the aHR and 95% CI. Then polynomial regression was used to fit the aHR, upper control limits, and lower control limits, separately. 

Due to the uneven sample sizes of the six groups, the 95% CI were narrow in the cDDD ≤ 20 and >100 groups, while they were much wider in the cDDD 40–60, 60–80, and 80–100 groups; however, the dose–response relationship was clear. For all-cause death, except for the cDDD ≤ 20 group, all the other dose subgroups were associated with a risk increase. The observed relationship was not linear and had a risk peak at cDDD 60–80, i.e., lowest at 20 cDDD, highest at 60 cDDD, and then decreased as the dose level further increased. (Appendix A). Results regarding cancer-specific death were similar to those for all-cause death (Appendix A). There were no obvious significant dose–response in the recurrence (Appendix A).

## 4. Discussion

In this large population-based matched-cohort study, the use of PPIs was associated with higher incidence per person–year and a higher adjusted hazard ratio for both all-cause death and cancer-specific death. The PPI exposure > 60 cDDD was associated with all-cause death and cancer-specific death. The dose–response relationship was not linear, and the peak of risk was at cumulative DDD 60–80 (Appendix A). PPI use was associated with a decreased risk of recurrence; however, there was no dose-dependent response between increased PPI exposure and reduced recurrence risk in the subgroup analysis. (Appendix A).

Multiple reports have indicated the tumor-suppressing effects of PPIs in several aspects. PPIs suppressed cell proliferation in CRC cell lines and carcinogenesis in a rat azoxymethane (AOM) model [28]. In cell-line and animal studies, it has been shown that pantoprazole inhibited the tumorigenesis and progression of CRC by suppressing T-cell-originated protein kinase [29]. Another identified pathway of PPIs’ antitumor effects was through the inhibition of membrane-bound ATP-binding cassette transporters and reduced drug resistance [30]. PPIs might enhance the efficacy and safety of anticancer agents via the off-target inhibition of supportive therapy during cancer chemotherapy [31]. In 2023, a systematic review based on 26 studies, concluded that PPIs might be beneficial as part of CRC therapy, in part due to their anti-tumor properties [32].

On the other hand, several hypotheses suggest that prolonged PPI use negatively interferes with CRC patient outcomes. First, PPI-induced hypergastrinemia, which has trophic effects on colonic mucosa and can stimulate carcinogenesis and cancer invasion [33]. Second, PPIs have been reported to influence colorectal cancer-cell-line survival in vitro, including promoting cell growth and metastasis [34]. Thirdly, gut microbiota overgrowth might be induced by decreased gastric acidity [35], dysbiosis, and increased nitrites and N-nitroso compounds, which are carcinogenic [36]. Moreover, dysbiosis caused by PPIs could lead to increased drug metabolism, altered autophagy, or immunosuppression [37]. In another retrospective chart review of 389 patients with stage II–III CRC treated with CapeOX or FOLFOX between 2004 and 2013, unadjusted analyses showed that PPI was associated with lower 3-year relapse-free survival in CapeOX-treated CRC patients, but not in FOLFOX-treated CRC patients. The association was not significant in overall survival [38].

In this study, the high-dose PPI exposure (>60 cDDD) was associated with increased risk of all-cause death and cancer-specific death. Although the dose–response relationship of PPI on all-cause death and CRC-specific death were not linear and had a risk peak at cDDD 60–80, it still indicated the trend that the increased PPI exposure was associated with an increased risk of death. Although PPI use was associated with reduced recurrence, there was no significant dose-dependent effect in recurrence reduction.

This study indicated that PPIs may play a complex role in CRC patients. The long-term alterations of physiological status, such hypergastrinemia or an altered microbiome maybe possibly be associated with complex effects in patients with CRC. Appendix A is a schematic representation of PPI effects in colorectal cancer.

### 4.1. Strengths of the Study

The current study possesses a number of strengths. First, it was a nationwide population-based study using a database for reimbursement purpose of medical expenses and a national cancer registration system with a large sample size and without loss of follow-up, which offered a good opportunity to explore the effect of PPIs on CRC patients. Second, this study used a person–years approach to determine incidence rate, reducing the bias from different observation time. Third, frequency matching was used to minimize the bias between these two groups. Fourth, for evaluating drug effect, the cases and controls were collected under new user design, and the observational period was well-designed.

### 4.2. Limitations of the Study

Several limitations exist in this study. First, the NHIRD-based studies did not include data on risk behaviors such as lifestyle or different dietary patterns. Second, NHIRD does not include data on over-the-counter use or patient compliance, since PPI exposure is measured by prescribed claims. Third, this is a retrospective cohort study, rather than a randomized study. Fourth, the more PPI use might be related to more comorbidities, but we tried to minimize the limitations by frequency matching. Since the study only included data from Taiwan’s population, it might be difficult to draw generalized conclusions in a continental or worldwide context.

## 5. Conclusions

The current results suggest that prolonged PPI use after CRC was associated with worse survival and this was associated with higher PPI exposure. The recurrence was reduced but there was no dose-dependent response. The potential risks and benefits of long-term maintenance of PPI therapy need to be carefully evaluated by physicians. Further randomized studies, or studies focusing on CRC subgroup patients, are needed to further comprehensively investigate the pros and cons of long-term PPI usage in CRC patients.

## Figures and Tables

**Figure 1 cancers-15-05304-f001:**
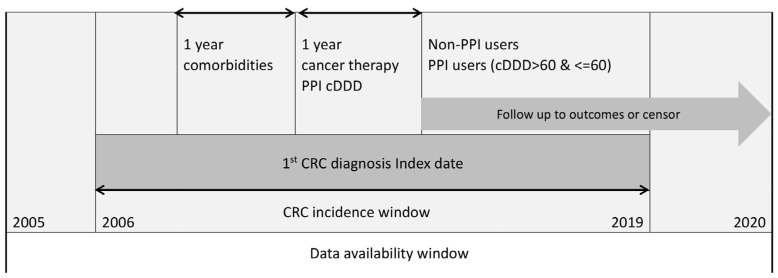
The study design.

**Figure 2 cancers-15-05304-f002:**
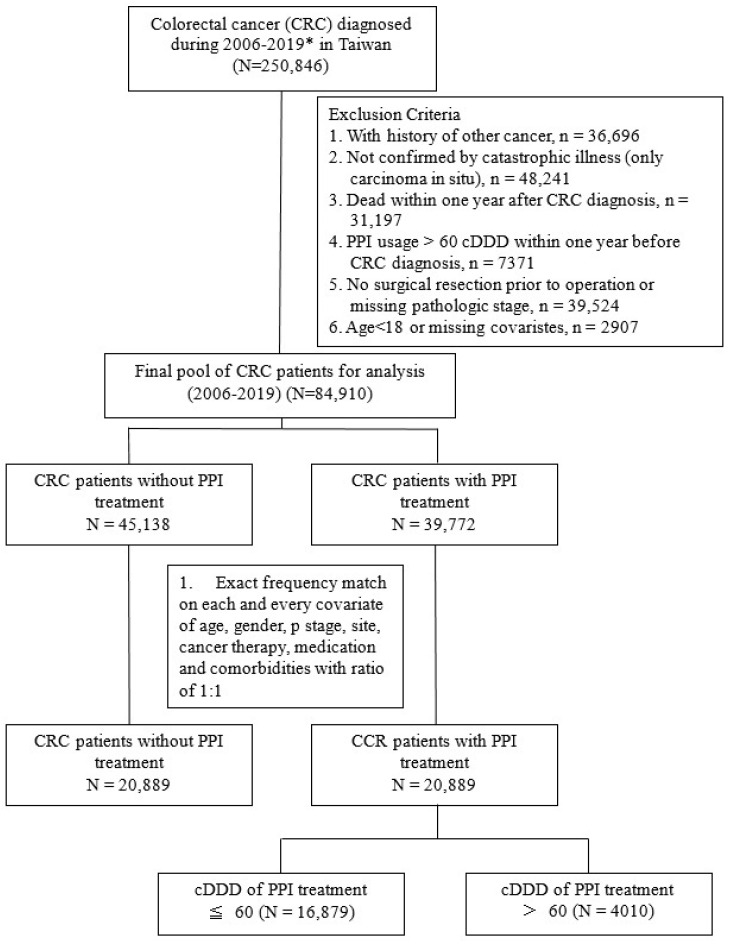
Study design flowchart of the cohort study. * Colorectal cancer (CRC) patients’ data between 2005 and 2020 in Taiwan was obtained from National Health Insurance Research Database and the Taiwan Cancer Registry. The comorbidities were evaluated based on the year 2005. CRC patients diagnosed between 2006 and 2019 were incorporated. Only newly diagnosed CRC patients were incorporated into this cohort study. All patients were followed up until the year 2020 or death, with at least a one-year follow-up period.

**Figure 3 cancers-15-05304-f003:**
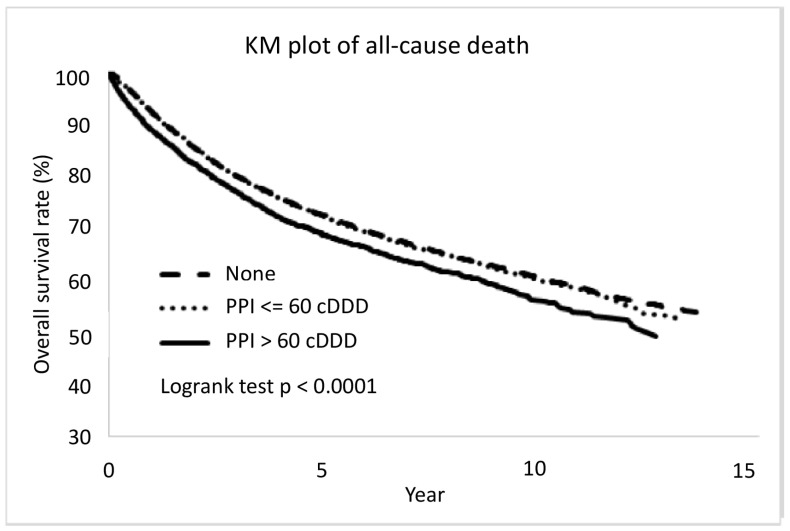
The CRC patients with proton pump inhibitor more than 60 cDDD were associated with more all-cause death. PPI: proton pump inhibitors; cDDD: cumulative defined daily dose; KM plot: Kaplan–Meier methodologies with log-rank test.

**Figure 4 cancers-15-05304-f004:**
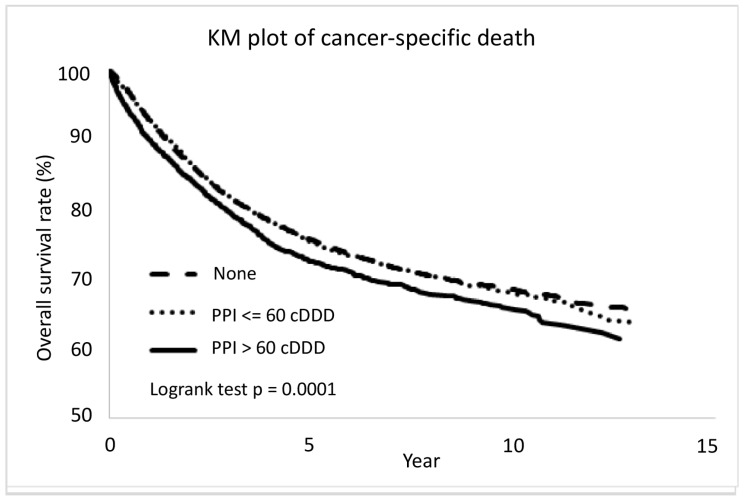
The CRC patients with proton pump inhibitor more than 60 cDDD were associated with more cancer-specific death. PPI: proton pump inhibitors; cDDD: cumulative defined daily dose; KM plot: Kaplan–Meier methodologies with log-rank test.

**Figure 5 cancers-15-05304-f005:**
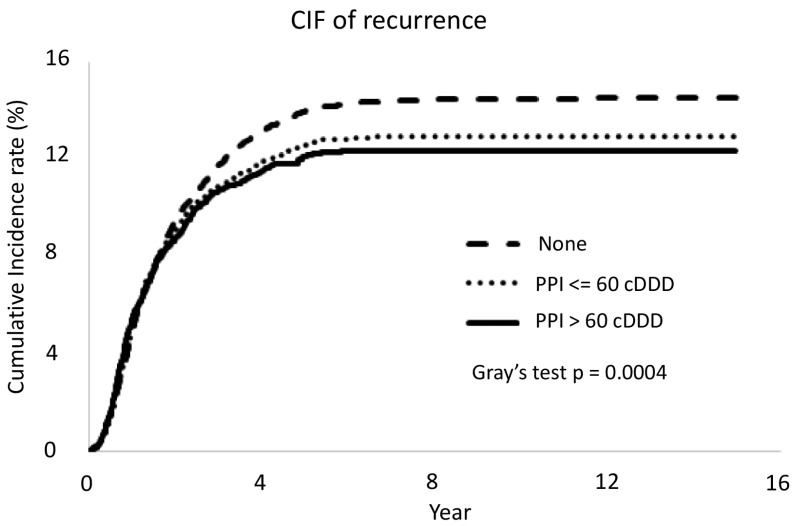
In the CRC patients with proton pump inhibitor, the recurrence was lower but no obvious dose-related effect was seen. CRC: colorectal cancer; PPI: proton pump inhibitor; cDDD: cumulative defined daily dose.

**Table 1 cancers-15-05304-t001:** Patients’ baseline characteristics after exact frequency match.

	PPI Non-Users	PPI Users	
Covariate	N (%), n = 20,889	N (%), n = 20,889	*p* Value
Age (year)			1.000
18–55	6541 (31.3)	6541 (31.3)	
56–65	6349 (30.4)	6349 (30.4)	
66–75	4765 (22.8)	4765 (22.8)	
>75	3234 (15.5)	3234 (15.5)	
Sex			1.000
Male	11,594 (55.5)	11,594 (55.5)	
Female	9295 (44.5)	9295 (44.5)	
Cancer site			1.000
Colon left	6190 (29.6)	6190 (29.6)	
Colon right	3834 (18.4)	3834 (18.4)	
Colon unspecified	4277 (20.5)	4277 (20.5)	
Rectum	6588 (31.5)	6588 (31.5)	
Pathological stage			1.000
I	4543 (21.8)	4543 (21.8)	
II	6118 (29.3)	6118 (29.3)	
III	7700 (36.9)	7700 (36.9)	
IV	2528 (12.1)	2528 (12.1)	
Cancer treatment			
Chemotherapy	9424 (45.1)	9424 (45.1)	1.000
Radiotherapy	2282 (10.9)	2282 (10.9)	1.000
Surgery	20,889 (100)	20,889 (100)	1.000
Targeted therapy	1853 (8.9)	1853 (8.9)	1.000
Comorbidity			
Coronary heart disease	942 (4.5)	942 (4.5)	1.000
Congestive heart failure	48 (0.2)	48 (0.2)	1.000
Asthma	181 (0.9)	181 (0.9)	1.000
Interstitial lung disease	(0)	(0)	1.000
COPD	206 (1)	206 (1)	1.000
Liver cirrhosis	14 (0.1)	14 (0.1)	1.000
Diabetes mellitus	2999 (14.4)	2999 (14.4)	1.000
CKD	169 (0.8)	169 (0.8)	1.000
Stroke	484 (2.3)	484 (2.3)	1.000
Dementia	68 (0.3)	68 (0.3)	1.000
Hypertension	6808 (32.6)	6808 (32.6)	1.000
PUD	2315 (11.1)	2315 (11.1)	1.000
GI bleeding	1533 (7.3)	1533 (7.3)	1.000
Medication			
NSAIDs usage	16,283 (78)	16,283 (78)	1.000
Steroids usage	13,635 (65.3)	13,635 (65.3)	1.000
CCI			1.000
0	12,548 (60.1)	12,548 (60.1)	
1–2	5807 (27.8)	5807 (27.8)	
≥3	2534 (12.1)	2534 (12.1)	
Urbanization			0.0002
High	5006 (24)	4669 (22.4)	
Median	10,284 (49.2)	10,396 (49.8)	
Low	5599 (26.8)	5824 (27.9)	
Region			0.0000
North	9140 (43.8)	8866 (42.4)	
Central	4678 (22.4)	5723 (27.4)	
East	343 (1.6)	478 (2.3)	
South	6728 (32.2)	5822 (27.9)	
SES (monthly income)			0.0226
≤20.1 K	7394 (35.4)	7118 (34.1)	
20.1–22.8 K	3049 (14.6)	3105 (14.9)	
22.8–42 K	5613 (26.9)	5641 (27)	
≥42 K	4833 (23.1)	5025 (24.1)	

PPI: proton pump inhibitor; COPD: Chronic obstructive pulmonary disease; CKD: chronic kidney disease; PUD: peptic ulcer disease; GI: gastrointestinal; NSAID: non-steroid anti-inflammation drug; CCI: Charlson comorbidity index; SES: socioeconomic status.

**Table 2 cancers-15-05304-t002:** Incidence proportion (IP) and event rate.

	All-Cause Death		CRC-Specific Death		Recurrence *	
	IP (95% CI)	Event (%)	IP (95% CI)	Event (%)	IP (95% CI)	Event (%)
PPI non-users	57.1 (55.7, 58.5)	6454 (30.9)	41.4 (40.2, 42.6)	4677 (22.4)	22.6 (21.8, 23.4)	2837 (13.6)
PPI users	63.1 (61.5, 64.8)	5520 (26.4)	46.8 (45.3, 48.2)	4087 (19.6)	23.9 (22.9,24.8)	2422 (11.6)
cDDD ≤ 60	61.2 (59.4, 63.1)	4361 (25.8)	45.5 (44.0, 47.1)	3243 (19.2)	23.9 (22.9,25.0)	1973 (11.7)
cDDD > 60	71.5 (67.5, 75.8)	1159 (28.9)	52.1 (48.6, 55.7)	844 (21.0)	23.6 (21.5,25.9)	449 (11.2)

CRC: colorectal cancer; PPI: proton pump inhibitor; cDDD: cumulative defined daily dose; IP: incidence proportion, per 1000 person-years. * Subgroup analysis of recurrence is based on only patients who initially received definite surgery.

**Table 3 cancers-15-05304-t003:** Adjusted hazard ratio of all-cause death, colorectal cancer-specific death, and recurrence in proton pump inhibitor non-users and users.

	All-Cause Death		Cancer-Specific Death		Recurrence *	
	Adjusted HR (95% CI)	*p* Value	Adjusted HR (95% CI)	*p* Value	Adjusted HR (95% CI)	*p* Value
PPI non-users	1		1		1	
PPI users	1.05 (1.02, 1.09)	0.0055	1.04 (1.00, 1.08)	0.0436	0.89 (0.84, 0.94)	<0.0001
cDDD ≤ 60	1.04 (1.00, 1.08)	0.0496	1.03 (0.99, 1.08)	0.1611	0.90 (0.85, 0.95)	0.0003
cDDD > 60	1.10 (1.04, 1.18)	0.0021	1.09 (1.01, 1.17)	0.0210	0.84 (0.76, 0.95)	0.0012

HR: hazard ratio; CI: confidence interval; PPI: proton pump inhibitor; cDDD: cumulative defined daily dose. * Subgroup analysis of recurrence is based on only patients who initial received definite surgery.

## Data Availability

The datasets generated and/or analyzed during the current study are not publicly available in accordance with the policy of the Health and Welfare Data Science Center, Ministry of Health and Welfare, Taiwan, but are available from the corresponding author upon reasonable request.

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
