# Peer review of "Long-Term Usage of Proton Pump Inhibitors Associated with Prognosis in Patients with Colorectal Cancer"

_cancers, 2023, doi:10.3390/cancers15215304_

Round 1

Reviewer 1 Report

Comments and Suggestions for Authors

The current manuscript on effect of PPI in colorectal cancer prognosis is really quite interesting, and its methodology seems robust. Hence, I only ask that the following changes are made before acceptance for publication:

- The authors should further comment on the study’s limitations, since it only included data from Taiwan population, and hence might be difficult to draw generalized conclusions, in a continental or worldwide context;

- The authors should avoid so many abbreviations, since it can become confusing for the reader; at least in the abstract abbreviations should be replaced by full writing;

- A schematic representation image of PPI effects (general, molecular, etc.) in colorectal cancer prognosis should be produced and added, for better reader visualization;

- In the ethics statement, when the authors say “This research was performed in accordance with the relevant guidelines and regulations.”, the mentioned guidelines and regulations should be added as references;

- Figures 2, 3, 4 and 5 quality (image resolution) should be improved;

- All tables should have the same format;

- An abbreviation list should be added.

Author Response

Dear Reviewer,

Please find enclosed our revised manuscript.We are grateful to the editors and peer reviewers for their thoughtful comments that helped us improve the quality of our manuscript. We answered all the reviewers’ comments and modified the manuscript accordingly.

Sincerely,

Wen-Yen Chiou (cwyncku@gmail.com)                    

Jui-Hsiu Tsai (dl07558@tzuchi.com.tw; faanvangogh@gmail.com)

The current manuscript on effect of PPI in colorectal cancer prognosis is really quite interesting, and its methodology seems robust. Hence, I only ask that the following changes are made before acceptance for publication:

- The authors should further comment on the study’s limitations, since it only included data from Taiwan population, and hence might be difficult to draw generalized conclusions, in a continental or worldwide context;

ANS: Thank you for your good comment. We added this sentence on our limitations. See it in detail on Page 13, lines 1-3.

… we tried to minimize the limitations by frequency matching. Since the study only included data from Taiwan population, and hence might be difficult to draw generalized conclusions, in a continental or worldwide context.

- The authors should avoid so many abbreviations, since it can become confusing for the reader; at least in the abstract abbreviations should be replaced by full writing;

ANS: The abbreviations in the abstract have been replaced by full writing. See it in detail on our Abstract on Page 1.

- A schematic representation image of PPI effects (general, molecular, etc.) in colorectal cancer prognosis should be produced and added, for better reader visualization;

ANS: Thank you for your valuable suggestion. We added a schematic representation image as supplementary figure 4 and noted in the part of Discussion on Page 12.

- In the ethics statement, when the authors say “This research was performed in accordance with the relevant guidelines and regulations.”, the mentioned guidelines and regulations should be added as references;

ANS: Thank you. We added 2 new references for the regulations and guidelines on the 2.1. Ethics statement and Reference sections on Pages 2, 14.

  1. Regulations on Human Trials. Ministry of Health and Welfare. 2016. Available online: https://law.moj.gov.tw/ENG/LawClass/LawAll.aspx?pcode=L0020162 (accessed on 30 October 2023).
  2. Personal Data Protection Act. National Development Council. 2023. Available online: https://law.moj.gov.tw/ENG/LawClass/LawAll.aspx?pcode=I0050021 (accessed on 30 October 2023).

- Figures 2, 3, 4 and 5 quality (image resolution) should be improved;

- All tables should have the same format;

ANS: The quality of figures is improved to 300 DPI. The format of all tables are uniformed.

- An abbreviation list should be added.

ANS: An abbreviation list is added as a supplementary text.

Reviewer 2 Report

Comments and Suggestions for Authors

As it is not obvious whether Proton Pump Inhibitors (PPIs)  inhibit or accelerate the course of CRC, and available data are inconclusive, presented paper sheds a new light on this very important issue. The manuscript was very good written and provides important information, also from practical, medical point of view. Below are some minor inaccuracies, which need to be corrected or explained: 

1. Presented manuscript does not have line numbering which makes this review hard to perform in terms of indicating some corrections. 

2. Introduction - "Several drugs have been identified as having the potential to influence CRC development and treatment, such as proton pump inhibitors(PPIs)[7], metformin[8], statin[9], or aspirin[10]." -Are these examples of drugs that both inhibit and accelerate the progression of the disease? This fragment is hard to understand. Please write it more clearly

3. In many European countries PPI drugs are OTC drugs, so everybody is able to buy it in pharmacy. I do not know what is the status of these drugs in Taiwan. The Authors wrote that "Patients who had more severe cancer status might lead to increased consumption of analgesics (including NSAIDs), thereby more PPI might be prescribed" - it follows that this type of medicines are available in Taiwan only with a doctor's prescription. If so, please state it clearly in the introduction. I have notice that it was explained in the chapter 4.2., but in my opinon it should be explained clearly at the beginning, as it make easy to understand the outcomes of the study. 

4. From the information provided by the Authors it is not obvious to me, which patients exactly were treated as PPI users? Patients who used PPIs before diagnosis of CRC or only patients who were started to take PPIs after CRC diagnosis? 

Author Response

Dear Reviewer,

Please find enclosed our revised manuscript.We are grateful to the editors and peer reviewers for their thoughtful comments that helped us improve the quality of our manuscript. We answered all the reviewers’ comments and modified the manuscript accordingly.

Sincerely,

Wen-Yen Chiou (cwyncku@gmail.com)                    

Jui-Hsiu Tsai (dl07558@tzuchi.com.tw; faanvangogh@gmail.com)

As it is not obvious whether Proton Pump Inhibitors (PPIs)  inhibit or accelerate the course of CRC, and available data are inconclusive, presented paper sheds a new light on this very important issue. The manuscript was very good written and provides important information, also from practical, medical point of view. Below are some minor inaccuracies, which need to be corrected or explained: 

  1. Presented manuscript does not have line numbering which makes this review hard to perform in terms of indicating some corrections.

ANS: I see!I agreed that no line numbering makes the review hard to perform it; but, we still thank you for reviewing it hard very much.

  1. Introduction - "Several drugs have been identified as having the potential to influence CRC development and treatment, such as proton pump inhibitors(PPIs)[7], metformin[8], statin[9], or aspirin[10]." -Are these examples of drugs that both inhibit and accelerate the progression of the disease? This fragment is hard to understand. Please write it more clearly

ANS: Thank you for your suggestion. We have tried to write if more clearly to the extent possible. We revised this sentence to “Several drugs have been identified as having the potential to influence reduce CRC development and treatment improved the prognosis, such as proton pump inhibitors(PPIs)[7], metformin[8], statin[9], or aspirin[10]; conversely, he association between long-term use of proton pump inhibitors(PPIs)  and increased CRC risk was also reported [7].” on Page 2, lines 6.

  1. In many European countries PPI drugs are OTC drugs, so everybody is able to buy it in pharmacy. I do not know what is the status of these drugs in Taiwan. The Authors wrote that "Patients who had more severe cancer status might lead to increased consumption of analgesics (including NSAIDs), thereby more PPI might be prescribed" - it follows that this type of medicines are available in Taiwan only with a doctor's prescription. If so, please state it clearly in the introduction. I have notice that it was explained in the chapter 4.2., but in my opinon it should be explained clearly at the beginning, as it makes easy to understand the outcomes of the study. 

ANS: Thank you for your suggestion.

(1). In Taiwan, PPI drugs are not OTC drugs. Almost PPI drugs were asked for prescribing by specialist doctors/gastroenterologists

(2). We added a sentence “Therefore, when CRC had more severe cancer status, they might lead to increased consumption of analgesics (including NSAIDs), thereby more PPI might be prescribed” in the Introduction on Page 2, lines 11.

  1. From the information provided by the Authors it is not obvious to me, which patients exactly were treated as PPI users? Patients who used PPIs before diagnosis of CRC or only patients who were started to take PPIs after CRC diagnosis? 

ANS: Thank you.

 (1). First, our enrolled patients were patients with CRC. Due to treatment or development of CRC, they needed to prescribe PPI because they had troublesome GI. For example, many patients with worsening development of CRC had often cancer pain and prescribing for analgesics (including NSAIDs). Since analgesics may lead to feel troublesome in GI, these patients were prescribed for PPI for their GI problems.

 (2). Only patients who were started to take PPIs after CRC diagnosis.

Reviewer 3 Report

Comments and Suggestions for Authors

This nationwide retrospective cohort study with a large sample size aimed to explore the dose-response effect of PPIs on CRC patients. Overall, it is well designed study, statistical methods are appropriate, and the obtained results bring some novelty to this research field. There are only few concerns in the manuscript that need to be clarified.

Firstly, the authors should further explain cumulative defined daily dose (cDDD) in the text. How was it calculated for every patient? Did it include all PPIs?

Table 1 should be improved technically (e.g., column width). More importantly, why is the sample size n=27,206? It is not clear since it should be 20,889.

Since Results and Discussion are written separately, Results should contain only obtained results without explanations and general statements (e.g., ‘PPIs are widely prescribed to CRC patients’ – percentages already show that, there is no need for explanation).

There are several technical issues:

There are some hyphens/dashes throughout the text that should be deleted.

Helicobacter should be with the capital H.

Second paragraph in Results: (65.3% and 78%).

KM plot should be defined in the text.

Comments on the Quality of English Language

No major issues were detected.

Author Response

Dear Reviewer,

Please find enclosed our revised manuscript. We are grateful to the editors and peer reviewers for their thoughtful comments that helped us improve the quality of our manuscript. We answered all the reviewers’ comments and modified the manuscript accordingly.

Sincerely,

Wen-Yen Chiou (cwyncku@gmail.com)                    

Jui-Hsiu Tsai (dl07558@tzuchi.com.tw; faanvangogh@gmail.com)

This nationwide retrospective cohort study with a large sample size aimed to explore the dose-response effect of PPIs on CRC patients. Overall, it is well designed study, statistical methods are appropriate, and the obtained results bring some novelty to this research field. There are only few concerns in the manuscript that need to be clarified.

Firstly, the authors should further explain cumulative defined daily dose (cDDD) in the text. How was it calculated for every patient? Did it include all PPIs?

ANS: Thank you for suggestions. We added this sentence and 1 new reference for further explaining cDDD on 2.3. Study population and PPI exposure section on Page 3, lines 21-23 and Page 13 for a new reference.

Figure 1. The study design.

For assessment of dose-response, each patient’s cumulative defined daily dose (cDDD) of all PPIs was calculated by the World Health Organization’s recommendation [26]. To explore the effect of PPIs…

  1. WHO Collaborating Center for Drugs Statistics Methodology . ATC Index with DDDs 2003. WHO; Oslo, Norway: 2003. Available online: https://www.whocc.no/atc_ddd_index/ (accessed on 30 October 2023).

Table 1 should be improved technically (e.g., column width). More importantly, why is the sample size n=27,206? It is not clear since it should be 20,889.

 ANS: Thank you. This is our mission; the sample size n = 20,889 was correct. So we revised it on Page 5.

Since Results and Discussion are written separately, Results should contain only obtained results without explanations and general statements (e.g., ‘PPIs are widely prescribed to CRC patients’ – percentages already show that, there is no need for explanation).

ANS: Thank you for valuable suggestion. We deleted the sentences about explanations and general statements. See it in detail on the 3. Results section of Page 7.

PPIs are widely prescribed to CRC patients. Nearly over half (46.8%) of CRC patients in the current study were prescribed a PPI in their 1st year after their first CRC diagnosis. After frequency matching, PPI users and non-PPI users were comparable for each confounding factor (Table 1). Age as a discrete variable was fairly evenly distributed by cut-points at 55, 65, and 75 years with the average age 65. Males outnumbered females by 10%. Rectal cancer counted 31.5%, while left-sided colon cancer and right-sided colon cancer counted 29.6% and 18.4%, respectively, and 20.5% was coded as unspecified. The majority of CRC was diagnosed at clinical stage III (36.9%). All the CRC patients received surgical resection prior to adjuvant treatment, followed by chemotherapy (45.1%), radiotherapy (10.9%), and targeted therapy (8.9%).

The most common comorbidity was hypertension (32.6%) and diabetes mellitus (14.4%) which were comparable to that of the same age group in Taiwan’s general population. Approximately 11.1% and 7.3% of patients had a history of peptic ulcer disease (PUD) and gastrointestinal (GI) bleeding, respectively. The steroids and NSAIDs use were matched exactly as well in two groups (both 65.3%). Urbanization, region, and socioeconomic status (monthly income) primarily reflected the distribution of the general population as well. These patients' variables were included to adjust for availability and quality of medical care in Taiwan (Table 1). Table 1 summaries the demographic characteristics and medical conditions in patients with colorectal cancers between PPI user and non-PPI use.

There are several technical issues:

There are some hyphens/dashes throughout the text that should be deleted.

  1. We revised all.

Helicobacter should be with the capital H.

  1. We revised all.

Second paragraph in Results: (65.3% and 78%).

ANS: OK, see it in detail on Page 7 line last 8.

… (PUD) and gastrointestinal (GI) bleeding, respectively. The steroids and NSAIDs use were matched exactly as well in two groups (both 65.3%). Urbanization, region, and

KM plot should be defined in the text. …

ANS: OK. We add the sentence “The Kaplan-Meier methodologies with log-rank test were to estimate and compare the all-cause death in patients with colorectal cancer between PPI use and non-PPI use” in 2.6. Statistical methods Section on Page 7, lines 20-22.

… methodologies with log-rank test and cumulative incidence of recurrence by CIF with Gray’s modified Chi-squared test. In addition, the Kaplan-Meier methodologies with log-rank test were to estimate and compare the all-cause death in patients with colorectal cancer between PPI use and non-PPI use. Adjusted hazard ratios were estimated using Cox’s proportional hazard model. All …